# Biodegradation of Polyethylene by *Enterobacter* sp. D1 from the Guts of Wax Moth *Galleria mellonella*

**DOI:** 10.3390/ijerph16111941

**Published:** 2019-05-31

**Authors:** Liu Ren, Lina Men, Zhiwei Zhang, Feifei Guan, Jian Tian, Bin Wang, Jihua Wang, Yuhong Zhang, Wei Zhang

**Affiliations:** 1College of Life Science and Technology, Harbin Normal University, No. 1 Shida Road., Limin Economic Development Zone, Harbin 150025, China; rllr1@outlook.com; 2Biotechnology Research Institute, Chinese Academy of Agricultural Sciences, No. 12 Zhongguancun South Street., Beijing 100081, China; guanfeifei@caas.cn (F.G.); tianjian@caas.cn (J.T.); zhangwei02@caas.cn (W.Z.); 3College of Forestry, Shanxi Agricultural University, Taigu 030801, China; linamen81@163.com (L.M.); zhiweizhang2012@163.com (Z.Z.); 4Institute of Soil Fertilizer and Agricultural Water Saving, Xinjiang Academy of Agricultural Sciences, Urumqi 830091, China; wbx_wm@126.com

**Keywords:** environmental impact, *Enterobacter* sp., plastic biodegradation, polyethylene, wax moth

## Abstract

Plastic polymers are widely used in agriculture, industry, and our daily life because of their convenient and economic properties. However, pollution caused by plastic polymers, especially polyethylene (PE), affects both animal and human health when they aggregate in the environment, as they are not easily degraded under natural conditions. In this study, *Enterobacter* sp. D1 was isolated from the guts of wax moth (*Galleria mellonella*). Microbial colonies formed around a PE film after 14 days of cultivation with D1. Roughness, depressions, and cracks were detected on the surface of the PE film by scanning electron microscopy (SEM) and atomic force microscopy (AFM). Fourier transform infrared spectroscopy (FTIR) showed the presence of carbonyl functional groups and ether groups on the PE film that was treated with D1. Liquid chromatography-tandem mass spectrometry (LC-MS) also revealed that the contents of certain alcohols, esters, and acids were increased as a result of the D1 treatment, indicating that oxidation reaction occurred on the surface of the PE film treated with D1 bacteria. These observations confirmed that D1 bacteria has an ability to degrade PE.

## 1. Introduction

Plastic polymers have advantages of ductility, durability, and low cost, and are commonly used in agricultural films and food packaging. However, plastic pollution poses a serious threat to animal and human health. Mircroplastics are of particular concern, as microplastics are deposited in aquatic environments, and microplastics ingested by seabirds or fish accumulate in their stomachs, which may cause death [1,2]. Furthermore, the potential accumulation of microplastics in the food chain eventually could have adverse effects on human health [3,4,5].

Incineration, landfilling, and recycling of plastic waste are costly and may cause secondary pollution [1,6]. The development of biodegradable plastics in recent years could slow down the accumulation of plastics in the environment, but fails to completely eliminate environmental pollution at the source [7,8]. Biodegradation, an eco-friendly method of degradation, is the process by which organic materials are decomposed or broken down into smaller compounds, including CO_2_ and H_2_O, by microbial action. The process of biodegradation can be divided into four stages: (a) cells grow firmly on the surface of the plastic material and produce hydrophilic groups; (b) long-chain hydrocarbons are oxidized or hydrolyzed into short chains by enzymes produced by microbial population, and a new aggregated bond is formed; (c) short-chain polymers are further broken down into fatty acids; (d) fatty acids are oxidized and decomposed into H_2_O, CO_2_, and humus [9,10].

There are increasing research interests in the biodegradation of plastic polymers. Polyethylene (PE), the most widely used plastic polymer, is a synthetic polymer of high molecular weight containing a structure of linear saturated hydrocarbon, which can be expressed as -[CH_2_-CH_2_]_n_- [11]. The demand for PE accounted for about 30% of total plastic polymers in 2017, and the annual global production of PE is approximately 140 million tons [12,13]. Since the early 1970s, researchers have investigated the biodegradation of PE and found certain PE-degrading strains, including *Bacillus*, *Pseudomonas*, *Staphylococcus*, *Streptococcus*, *Streptomyces*, *Brevibacterium*, *Nocardia*, *Moraxella*, *Penicillium*, and *Aspergillus* from soil, marine, and sludge under natural conditions [1,13,14,15]. However, the strong hydrophobicity, high chemical bond energy, and high molecular weight of PE hinder its efficient degradation by most strains, especially within a short period of time [16]. It has been shown that the degradation of PE by fungus *Penicillium simplicissimum* and *Nocardia asteroides* could take several months or even longer [17]. Recently, Yang Jun et al. reported that PE could be significantly degraded by microorganisms of the Indian meal moths, and two strains, *Enterobacter asburiae* YT1 and *Bacillus* sp. YP1, were isolated. Following a 60-day incubation, approximately 6% and 11% of a PE film was degraded by YT1 and YP1, respectively [11]. These results indicate that insects could be a promising source to obtain PE-degrading microorganisms. Similarly, Paolo Bombelli et al. found that there was 92 mg mass loss of a PE shopping bag after exposure to ~100 wax worms, and ethylene glycol was produced for 12 hours [18]. Nonetheless, further studies are still needed to identify specific microorganisms that play a key role in the degradation of PE. Therefore, the aim of this study was to screen PE-degrading microorganisms from the guts of wax moth (*Galleria mellonella*), and the degradation efficiency and mechanisms were further determined using scanning electron microscopy (SEM) coupled to an energy dispersive spectroscopy (EDS), atomic force microscopy (AFM), Fourier transform infrared spectroscopy (FTIR), and liquid chromatography-tandem mass spectrometry (LC-MS).

## 2. Materials and Methods

### 2.1. PE Film and Wax Moth Larvae

PE film was purchased from the SINOPEC Beijing Yanshan Company in Beijing, China. The PE film was cut into a 40 mm × 40 mm square, disinfected with 75% ethanol and air-dried, and then used as the sole carbon source for the growth of microorganisms in a shake flask or on a plate.

The wax moth larvae were collected from natural bee farms in Beijing and raised at Shanxi Agricultural University (with beeswax as the main food). The beeswax, as a source of food for the wax moth, is a ‘natural plastic’ with a chemical structure similar to that of PE. Hence, the wax moth can be used to screen the microorganisms that “feed” PE plastic.

### 2.2. Medium

The liquid basal medium in which PE was the sole carbon source (LPEM) contained (per 1000 mL): 0.7 g of KH_2_PO_4_, 0.7 g of K_2_HPO_4_, 0.7 g of MgSO_4_·7H_2_O, 1.0 g of NH_4_NO_3_, 0.005 g of NaCl, 0.002 g of FeSO_4_·7H_2_O, 0.002 g of ZnSO_4_·7H_2_O, and 0.001 g of MnSO_4_·H_2_O, according to the ASTM standard (ASTM G22-76) [19]. A carbon-free source agar solid medium (APEM) was prepared by adding 15 g agar to 1000 mL of liquid basal medium. The liquid nutrient broth (LNB) medium was prepared by dissolving 3 g of beef extract, 10 g of bacteriological tryptone, and 5 g of NaCl in 1000 mL of deionized water and then the pH was adjusted to approximately 7.0. The solid nutrient broth (SNB) was prepared by adding 15 g of agar to 1000 mL of LNB. Physiological saline was prepared by dissolving 8.5 g of NaCl in 1000 mL of deionized water, and then the pH was adjusted to about 7.0. All media were autoclaved at 121 °C for 20 min.

### 2.3. Microbial Sample

Wax moth larvae were immersed in 75% ethanol for about 1 min for disinfection, and then the larvae’s guts were dissected and placed in a 50-mL centrifuge tube and washed with sterile physiological saline three times. Then, 40 mL of sterile physiological saline was added followed by vortexing to obtain an intestinal homogenate. The intestinal homogenate was used to screen PE-degrading microorganisms.

### 2.4. Screening of PE-Degrading Strains

The intestinal homogenate was inoculated into the LPEM (containing 1% PE) and cultured at 37 °C (220 r/min). After 31 days, the bacterial solution was coated on the SNB medium and cultured for 12 h. The colonies grown on the plate were considered as primary screening colonies for PE-degrading bacteria. 

### 2.5. The Biodegradation Test and Identification of PE-Degrading Strains

The primary screening colonies were inoculated into the LNB for 12 h. The cells were collected by centrifugation at 12,000 r/min and rinsed with sterile physiological saline, which was repeated three times to remove residual medium. Then, the cells were re-suspended with sterile physiological saline. The obtained suspension was inoculated into the LPEM, and a 5% PE sheet (40 mm × 40 mm) was added. Simultaneously, 500 uL of suspension was coated on the APEM and covered with the PE sheet (40 mm × 40 mm). A blank control, without inoculation of bacterial liquid, was also tested. There were three replicates for each sample. All the shake flasks and solid medium were kept at 37 °C (220 r/min) for 31 days. The optical density at 600 nm (*OD*_600_) of the shake flask was monitored regularly during the period of cultivation.

The genomic DNA was extracted from the bacterial solution by proteinase K treatment and phenol-chloroform. The 16S rDNA sequence was amplified using universal primers 27F (5′- AGAGTTTGATCMTGGCTCAG-3′) and 1492R (5′- GGTTACCTTGTTACGACTT-3′). The obtained sequences were submitted to the National Centre for Biotechnology Information (NCBI) GenBank database and aligned using the search tool Basic Local Alignment Search Tool (BLAST, http://www.ncbi.nlm.nih.gov/BLAST/).

### 2.6. Observation of SEM and AFM

The PE film was recovered from the shake flask after the cultivation, and then placed in the centrifuge tube containing sterile water and shaken on a vortex mixer according to the methods of Yang Jun et al. [11]. The dried PE film was cut into 5-mm sized pieces and coated with gold. The surface topography, biofilm, and atom contents of the microbe-treated or untreated PE were observed under SEM (Carl Zeiss NTS GmbH, Jena, Germany) coupled with EDS (Oxford Instruments, Abingdon, Oxfordshire, UK). The recovered PE film was washed with 2% w/v aqueous sodium dodecyl sulfate (SDS) to thoroughly remove the biofilm adhered to the PE surface and dried overnight [20]. The surface topography was observed by AFM (Dimension Icon, Veeco, Billerica, MA, USA) at a scan speed of 1.0 Hz.

### 2.7. Analysis of Spectroscopy

The characterization of functional groups of the PE film surface was determined by FTIR (Nicolet iN10MX, Nicolet, Madison, WI, USA). The view field of the FTIR microscope was 400 μm × 400 μm, and the scanning was performed at a step size of 10 μm under a contact pressure of 3 MPa. The absorbance ranged from 4000 cm^−1^ to 650 cm^−1^ with a scan resolution of 4 cm^−1^ for the FTIR. A background scan was performed each time before the sample was scanned, and the final sample spectrum was subtracted from the background scan value.

### 2.8. Detection of Water-Soluble Products

The 31-day culture solution from the shake flask was centrifuged at 12,000 r/min for 10 min to obtain the supernatant. The 30 mL supernatant was freeze-dried and then re-dissolved in 1 mL of 50% ethanol, and centrifuged again at 12,000 r/min for 10 min after performing ultrasound for 10 min. Then, 2 μL of supernatant was used for LC-MS (AB Sciex TripleTOF^®^ 5600+, AB SCIEX, Redwood, CA, USA) analysis. The degradation products were detected by LC-MS equipped with a Waters^TM^ HSS T3 (150 × 3 mm, 1.8 µm, Waters, Milford, MA, USA) at 100–1500 m/z. The column temperature was 35 °C and the flow rate was 0.300 mL/min during operation.

### 2.9. Statistical Analyses

All experiments were done in triplicate, and standard deviations have been presented as the error bars in Figure 1a and Figure 2a. The mean variables and standard deviation were analyzed by Statistical Program for Social Sciences (SPSS 20.0, Chicago, IL, USA). 

## 3. Results and Discussion

### 3.1. Screening of PE-Degrading Strains

Following a 31-day cultivation, four strains were screened that could grow in the LPEM containing 5% PE (Appendix A), and one strain, D1, was specifically selected because it had a better growth trend when PE was the sole carbon source. The growth of strain D1was analyzed based on the changes of *OD*_600_, as shown in Figure 1a. When strain D1 was grown for 7 days in the LPEM (containing 5% PE), the *OD*_600_ of the bacteria solution reached the highest value (0.24) and remained constant thereafter, whereas the *OD*_600_ of the control (without bacteria D1) did not change over the course of 31 days. Accordingly, at the end of the cultivation, the bacterial fluid inoculated with D1 was more turbid than the control shake flask (Figure 1b). Additionally, the colonization of D1 was observed around the PE film on the APEM on day 14 of the cultivation (Figure 1c,d).

When the degradation test was cultured for 14 days, the colonization of D1 was observed around the PE film on the APEM. This results was similar to that observed by Yang et al. [21]. The observations of turbid bacterial liquid and formed bacterial colonies confirmed the utilization of carbon sources by strain D1. Additionally, steady changes of *OD*_600_ were observed when the degradation test was cultured for 31 days. It could be inferred that D1 adapted to the nutrient conditions of the medium during the period of cultivation. Microbial adaptation to PE is a key factor for biodegradation. The present study showed that the amounts of D1 bacteria adhered to the PE film increased gradually with the extension of cultivation. Hence, D1 could potentially be effective in degrading PE. Sequence alignment of BLAST showed that the strain D1 could be *Enterobacter* sp. (GenBank accession no. MK934326). 

The degradation of PE may be affected by the interactions between microorganisms. The commercial PE could be degraded by *Bacillus licheniformis* and *Lysinibacillus bacterium* simultaneously [22]. The mixture of *Citrobacter* sp. and *Kosakonia* sp. was capable of degrading PE and polystyrene (PS) [23]. Hence, we are also considering adding other degradation strains for degradation test.

### 3.2. Determination of the Degradation Effect

The degradation characteristics of PE are usually determined by thermogravimetric analyzer (TGA), X-ray diffraction (XRD), gas chromatograph/mass spectrometer (GC/MS), SEM, AFM, and FTIR [24]. Generally speaking, determination of weight loss is a relatively simple method used to detect the degradation of PE, however, it may not be sensitive enough under the conditions of long periods of incubation and slow biodegradation rates [25]. Therefore, no weight loss test was performed in this study.

#### 3.2.1. Surface Micromorphology and Atomic Percentage

After a 31-day incubation, surface morphology and structural changes of the PE film were observed by SEM and AFM. Figure 2 shows that the surface morphology of PE film was changed by D1, but the surface of the control remained smooth and no microorganisms were observed (Figure 2b). Figure 2c shows that D1 adhered to the surface of the PE film during the growing period. The microbial morphology was more clearly seen under a high magnification (Figure 2d). An incomplete biofilm of D1 colony formed on the surface of PE after 31 days of cultivation (Figure 2d). Analysis of elemental changes on the PE surface by SEM coupled with EDS (SEM-EDS) showed that the percentage of oxygen atoms in the D1 group was higher than that of the control group after an incubation of 31 days; the percentage of carbon atom mass was reduced by 1.98%, and the percentage of oxygen atom mass was increased by 1.98% when compared with the control (Figure 2a, Appendix A). These results indicated that oxidation reaction occurred on the PE surface. Compared with the control group (Figure 3a), obvious depressions appeared on the surface of the PE film in the D1 group (Figure 3b), and the surface of the PE film was eroded by D1. In addition, cracks appeared on the microbial-treated PE film (Figure 2b), which might be related to the changes in the physical structure of the PE film during the period of biodegradation.

The results of SEM and AFM supported the findings, including the effect of D1 on the PE film from the perspective of microbial morphology, and the roughness, depressions, and cracks appeared on the PE surface. Notably, the biofilm was observed on the surface of PE film. It has been shown that microbial attachment and biofilm formation facilitated the contact of PE with microbial enzymes secreted by D1, thereby making the long carbon-carbon bond more susceptible to oxidation [20,23,26]. Tribedi and Sil reported the formation of biofilm by *Pseudomonas* sp. AKS2 strain during the degradation of polyethylene [27]. The stable AKS2 biofilm highly enhanced the hydrophobicity level of a PE film, which accelerated the degradation rate of PE [28]. 

It is indeed useful to see the progression of the bacterial growth and monitor the structure of PE at different times [29,30]. Based on the current findings, we will observe the surface morphology of PE films at different culture times to further analyze how the biofilm is formed and changed; also, we will screen and verify the degradation of PE by other strains from the gut of wax moth in future works.

#### 3.2.2. Changes in Chemical Structure of the PE Film Surface

FTIR was used to analyze chemical compositions of the PE surface to help evaluate the occurrence of biodegradation. The infrared spectrum of the PE surface inoculated with D1 for 31 days showed an absorption peak at 1652 cm^−1^ and 1075 cm^−1^ corresponding to carbonyl groups (-C=O) and ether groups (-C-O-C-), respectively (Figure 4). Additionally, the absorption peaks at 730 cm^−1^, 1435.78 cm^−1^, and 1450 cm^−1^ were observed regardless of whether the PE surface was treated with strain D1 or not. The presence of carbonyl and ether group indicated the cleavage or formation of new bonds that could promote the oxidation of PE. It was previously reported that the appearance of carbonyl groups on the spectrum is an indication of the biodegradation of PE [24]. It is possible that the chemical structure of PE was weakened because of the skeletal vibration during the cultivation, based on the results that both the treated group and control group had an intensive absorbance at around 730 cm^−1^. The appearance of bands approximately at 1450 cm^−1^ was due to C-H bending vibrations in the long-chain backbone of PE. 

A hydrogen atom on a long carbon-carbon bond might be replaced by an oxygen atom, and functional groups such as a carbonyl group, an ester group, or an ether group formed. These functional groups contain oxygen when the PE is oxidized. Reports have also shown that some classical groups such as C-H groups, -C=O groups, -C-O-O groups, and -CH_2_ groups generally correspond to absorption peaks at 3000–2840 cm^−1^, 1730–1650 cm^−1^, 1150–1075 cm^−1^, and 900–735 cm^−1^, respectively [17,31,32]. Increases in ketones and double bonds provide evidence of polyethylene biodegradation, according to Balasubramanian et al. [33]. In the present study, the detection of carbonyl and ether on the PE film incubated with D1 proved that the oxidation reaction occurred. It has been shown that oxidation of PE enhanced hydrophilicity and ultimately facilitated PE biodegradation [34,35,36]. Some pretreatments, including photo-oxidation, thermal treatment, and acid treatment, have been shown to accelerate the oxidation and degradation of PE [37].

#### 3.2.3. Analysis of Water-Soluble Products

The water-soluble products of the solution resulting from the 31 days of cultivation of the PE film were analyzed by LC-MS. The eluted compounds contained almost all C, H, and O elements. Significant differences were found in the abundance, suggesting the compounds were different between the two groups (Figure 5a). The compounds including alcohols, esters, and acids were significantly increased in the D1-treated group compared with the control group (Appendix A). The content of ethyldodecanoate and 6-methyl-5-hepten-2-ol in the D1 group was 4 times higher than the control group (Figure 5b). Several compounds such as monobenzyl phthalate and N-Acetylglutamic acid were detected and considered as newly formed compounds in the D1 group, while they were not detected in the control group (Figure 5b). 

Alcohols, alkanes, hydrocarbon, esters, and acids are detected by LC-MS, which reflects the metabolism of bacteria during the PE biodegradation [38]. Monobenzyl phthalate is the main metabolite of butyl benzyl phthalate (BBP). We speculated that the production of monobenzyl phthalate was caused by some oxidoreductases secreted by the strain *Enterobacter* sp. and the specific oxidation mechanism was further studied. The current study showed that the long-chain carbon-carbon bond of PE was oxidized during the period of cultivation, given the results showing that alcohols, esters, and acids were significantly increased. Although the control and D1-treated group shared some compounds, their contents were significantly different. The differences were attributed to the utilization of nitrogen and carbon sources by microorganisms in the medium and the photooxidation of PE film occurred during the cultivation [21,25,38]. Furthermore, some other compounds such as N-acetylglutamic acid might be the secondary metabolic products secreted by the strain D1 when utilizing PE to grow. These differences in the compounds could help demonstrate that PE was degraded by strain D1 [21]. 

Currently, we have initially detected the production of alcohols, alkanes, hydrocarbon, esters, and acids. The biodegradation mechanism of PE is complex and involves the participation of various oxidoreductases. It has been reported that the biodegradation of PE by microbial enzymes is much more efficient relative to microorganisms [39]. Studies have shown that extracellular enzymes were capable of attacking and modifying PE films and that the laccase-mediator system could decrease the *M*w of polyethylene from 242,000 to 28,300 for 3 days [40,41]. However, the degrading enzymes that have been studied usually have poor stability, and the mechanism of biodegradation of PE remains limited in the literature. Therefore, it would be of great interest to identify enzymes from microorganisms that can efficiently degrade PE, and investigate possible mechanisms underlying the enzymatic degradation.

## 4. Conclusions and Future Perspectives

In this study, the strain *Enterobacter* sp. D1 was screened from the gut homogenate of the wax moth by using PE as the sole carbon source. Further analysis confirmed that PE could be degraded by strain D1. As a carbon source, PE materials are not easily utilized by microorganisms compared to glucose and beef extract, due to their structural particularities (hydrophobicity, high molecular weight, etc.). The current research shows that the desirable degradation effect of PE was not achieved at the laboratory level regardless of microbes or degrading enzymes. This is a challenge for us to continue to study the degradation effect of *Enterobacter* sp. on PE. 

In our future studies, we are planning to use the following methods to potentially improve the degradation rate: (1) addition of other PE degradation strains to the culture medium for possible synergistic effects; (2) addition of surfactants (such as Tween-80, mineral oil, or paraffin oil) to the culture medium to potentially improve the hydrophilicity of the PE surface and accelerate the formation of biofilm; (3) pretreatment (ultraviolet irradiation, acid treatment, etc.) of the PE film; and (4) mutation and domestication of the degrading strains.

## Figures and Tables

**Figure 1 ijerph-16-01941-f001:**
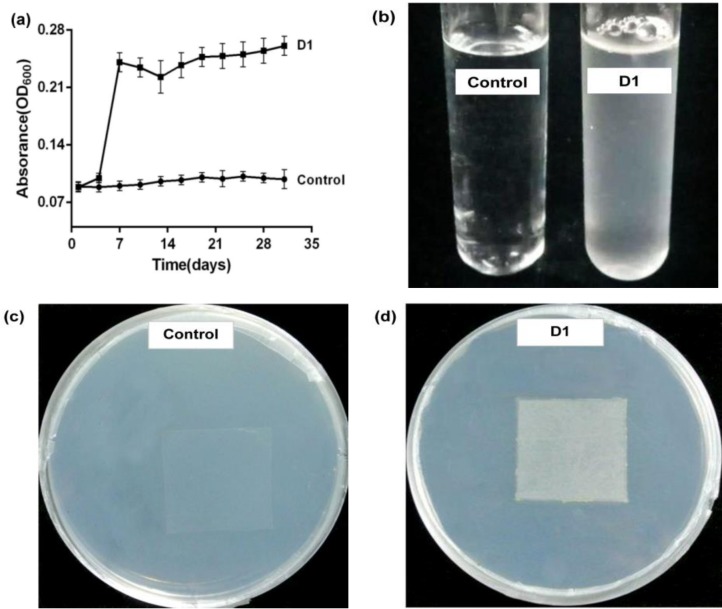
The growth of bacteria D1 in the medium in which a polyethylene (PE) film was the only carbon source. (**a**) The changes of optical density at 600 nm (*OD*_600_) during the 31-day cultivation. (**b**) Turbidity of the D1 bacteria solution. (**c**) Plate of control group containing PE film without D1. (**d**) D1 colonies grown around the PE film on the carbon-free source agar solid medium (APEM).

**Figure 2 ijerph-16-01941-f002:**
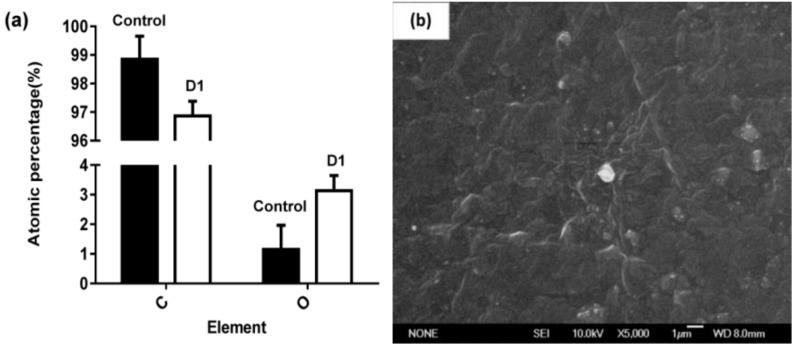
SEM photographs of PE film untreated and treated with D1 and analysis of atom contents on the PE film. (**a**) Changes in the percentage of carbon and oxygen atoms on the PE film surface after a 31-day incubation. (**b**) The control group without D1 bacteria (5000×). (**c**) The D1 on the PE surface after 31 days of cultivation under a low magnification lens (5000×). (**d**) The D1 on the PE surface after 31 days of cultivation under a high magnification lens (20000×).

**Figure 3 ijerph-16-01941-f003:**
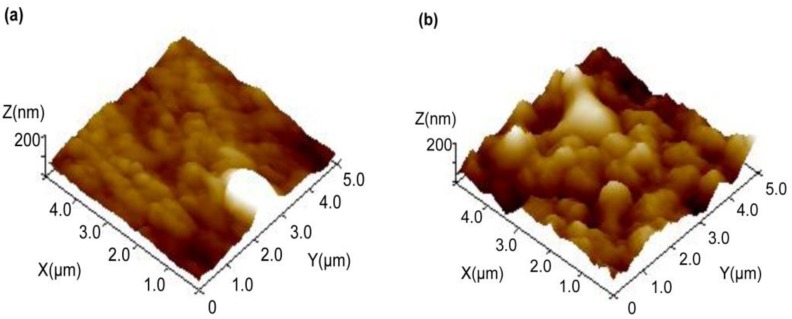
Physical topography of PE film untreated (**a**) and treated with D1 bacteria (**b**) by atomic force microscopy (AFM).

**Figure 4 ijerph-16-01941-f004:**
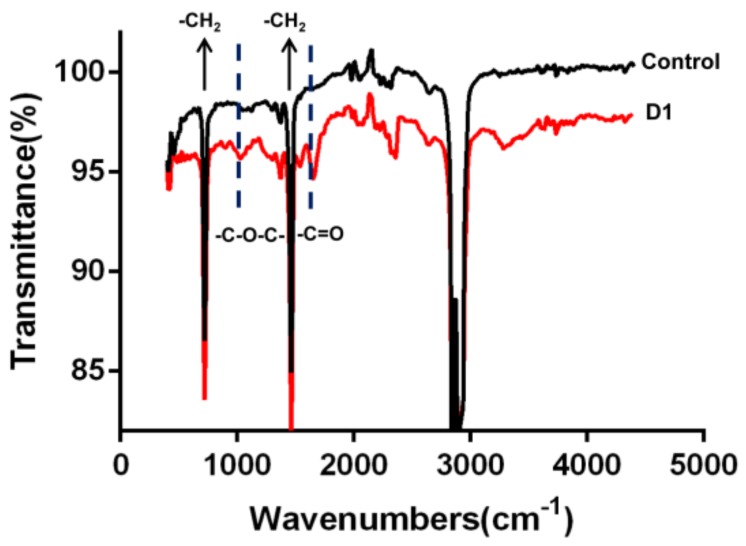
Distribution of carbonyl bands (-C=O, 1652 cm^−1^) and ether groups (-C-O-C-, 1075 cm^−1^), observed with a FTIR microscope on the PE film treated with D1 for 31 days.

**Figure 5 ijerph-16-01941-f005:**
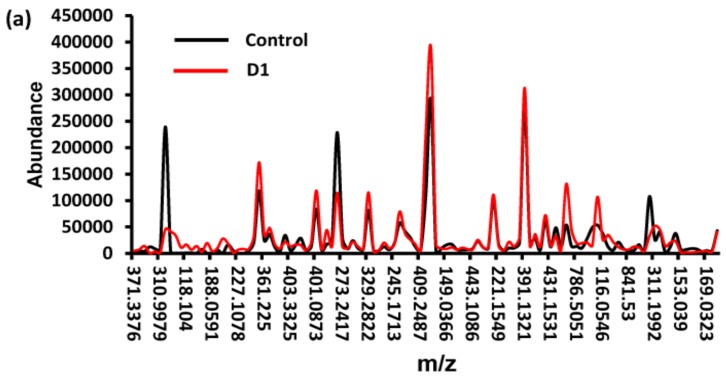
Detection of water-soluble products by LC-MS at the end of 31 days of cultivation. (**a**) Comparison of the abundance of compounds eluted at the same M/Z between the control group and the D1-treated group. (**b**) Increased abundances of acids, esters, and alcohols were observed in the D1-treated group compared with the control group.

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
