# Peer review of "Biodegradation of Polyethylene by Enterobacter sp. D1 from the Guts of Wax Moth Galleria mellonella"

_ijerph, 2019, doi:10.3390/ijerph16111941_

Reviewer 1 Report

The authors in this paper are discussing the results of biodegradation of plastics. The authors were able to show, by various analyses, the biodegradation of PE occurs after 31 days. I believe the applications of their work are immense and highly significant if their conclusions are accurate. The text is well-written and the methods are explained well enough. Here are my comments:

Why was 31 days chosen as the duration of the experiment?

It looks like triplicates were done for all experiments, what was the standard deviation? Also, which sample is presented in Figure 2? Do all samples look the same?

I think it would be useful to see the progression of the bacterial growth with time. Would it be possible to observe the samples at different times?

The bacteria should grow exponentially, assuming there is no limitation in nutrient supply, with the availability of substrate. This means that the bacteria should degrade the plastics faster with time as long as they have enough nutrients. If we could see only small fraction of biodegradation in 31 days, I am not sure if we will further decay of plastics.

Based on the data collected, what is the rate of biodegradation? How many years would it take for the sample to be completely biodegradated by the bacteria?

Would you be able to scale this process? How would you increase the rate?

I am not entirely sure if we can make conclusive statements with the data that is presented. I agree that there was some biodegradation and change in the molecular structure based on the results presented after 31 days. However, this is not conclusive evidence that bacteria will continue to biodegrade the plastics after 31 days. Without being able to say that the bacteria will completely biodegrade the plastics within a reasonable time-frame, these results have no context. I would recommend the authors to remove the broad claims about the biodegradation of plastics. I also recommend rejecting the paper or major revision by better contextualizing these results.

Author Response

Response to Reviewer 1 Comments

Point 1: Why was 31 days chosen as the duration of the experiment?

Response 1: The literature indicate that microbial degradation of polyethylene (PE) is a relatively slow process under laboratory conditions. It is possible to observe the increases in ether and carboxyl functional groups when PE film is incubated with strain Stenotrophomonas pavanii (CC18) for 56 d [1], and it is possible that the properties and morphology of PE bags significantly changed following the treatment by Bacillus sp. BCBT21 for 30 d [2]. The biodegradation process is affected by many factors including microbial species, medium composition, culture environment and structure of PE, therefore, we cannot predict whether the strain Enterobacter sp. D1 would degrade PE film before conducting the experiment. The selection of the 31-d duration of the experiment was mainly based on the previous study that the Enterobacter asburiae YT1 could cause obvious damage to the PE films over a 28-d incubation [3], and the steady changes of OD600 were observed when it was cultured for 31 days, along with the results that the physical and chemical properties of the PE film changed by the analysis of SEMAFM and FTIR in the current study. Certainly, the similar results may also be observed if the period of cultivation is 28 d or 35 d.

Point 2: It looks like triplicates were done for all experiments, what was the standard deviation? Also, which sample is presented in Figure 2? Do all samples look the same?

Response 2: All experiments were done in triplicate, and standard deviation of the OD600 and atom percentage have been presented as the error bar in Figure 1a and Figure 2a in the manuscript, respectively. The specific values are as follows:

Table 1 The means and standard deviation values of the samples in Figure 1a of the revised manuscript

Control

D1

Days

Mean

SD

Mean

SD

1

0.089  

0.006  

0.089  

0.006  

4

0.089  

0.006  

0.100  

0.006  

7

0.090  

0.006  

0.241  

0.012  

10

0.092  

0.006  

0.234  

0.012  

13

0.096  

0.006  

0.223  

0.020  

16

0.097  

0.006  

0.237  

0.015  

19

0.101  

0.006  

0.247  

0.012  

22

0.099  

0.010  

0.248  

0.015  

25

0.102  

0.006  

0.250  

0.015  

28

0.100  

0.006  

0.255  

0.015  

31

0.098  

0.012  

0.261  

0.012  

Table 2 The means and standard deviation values of the samples in Figure 2a of the revised manuscript

Atom percentage

Sample

Mean

SD

%C

Control

96.867

0.103

D1

98.847

0.163

%O

Control

3.133

0.103

D1

1.153

0.163

Analysis of physical morphology and chemical structure of PE film were done after completing the 31-day degradation test. The triplicate samples of the treated or untreated PE film showed the same results, therefore, one of them was presented in Figure 2.

Point 3: I think it would be useful to see the progression of the bacterial growth with time. Would it be possible to observe the samples at different times?

Response 3: We highly thank the reviewer for the suggestion. It is indeed useful to see the progression of the bacterial growth and monior the structure of PE at different times [4-6]. In this study, our aim was to screen microorganisms that are capable of degrading polyethylene from the gut of wax moth larvae, so the endpoint degradation test was mainly focused. However, based on the current findings that strain Enterobacter sp. D1 is capable of degrading PE isolated from the gut of wax moth larvae, we will observe the surface morphology of PE film at different culture times to further analyze how the biofilm is formed and changed; also, we will screen and verify the degradation of PE by other strains from the gut of wax moth in the future work.

Point 4: The bacteria should grow exponentially, assuming there is no limitation in nutrient supply, with the availability of substrate. This means that the bacteria should degrade the plastics faster with time as long as they have enough nutrients. If we could see only small fraction of biodegradation in 31 days, I am not sure if we will further decay of plastics.

Response 4: As a carbon source, PE materials are not easily utilized by microorganisms compared to glucose and beef extract, due to their its structural particularity (hydrophobicity, high molecular weight, etc.). The current research shows that the desirable degradation effect of PE was not achieved at the laboratory level regardless of microbes or degrading enzymes. This is also a challenge for us to continue to study the degradation effect of Enterobacter sp. on PE. However, we also unexpectedly found that when the microorganisms were cultured with PE for a longer period of time (several months), the surface state of the PE film greatly changed (data not shown), and specific verification and analysis over a long time are still needed.

Point 5: Based on the data collected, what is the rate of biodegradation? How many years would it take for the sample to be completely biodegradated by the bacteria?

Response 5: The mass of PE film is approximately about 10.3% by the determination of gravimetric method before and after cultivation. Generally speaking, determination of weight loss is a relatively simple method used to detect the degradation of PE, however, it may not be sensitive enough under the conditions of long period of incubation and slow biodegradation [6]. Hence, the other methods for determining degradation rate will be performed.

The high molecular weight and strong hydrophobicity of PE makes it difficult to be degraded. The PE waste enters into the landfills and takes hundreds of years to be degraded in the natural environment. Due to the preference of microorganisms for natural nitrogen sources and carbon sources in the natural environment, the degradation rate of PE would be faster if the degradation test is carried out under laboratory conditions, compared with the natural environment.

Point 6: Would you be able to scale this process? How would you increase the rate?

Response 6: At present, no amplification experiments have been carried out, but we are using the following methods to improve the degradation rate: 1) addition of other PE degradation strains to the culture medium for synergistic effect; 2) addition of surfactants (such as Tween-80, mineral oil or paraffin oil) to the culture medium to improve the hydrophilicity of the PE surface and accelerate the formation of biofilm; 3) pretreatment (ultraviolet irradiation, acid treatment, etc) of PE film4) mutation and domestication of the degrading strains.

Point 7: I am not entirely sure if we can make conclusive statements with the data that is presented. I agree that there was some biodegradation and change in the molecular structure based on the results presented after 31 days. However, this is not conclusive evidence that bacteria will continue to biodegrade the plastics after 31 days. Without being able to say that the bacteria will completely biodegrade the plastics within a reasonable time-frame, these results have no context. I would recommend the authors to remove the broad claims about the biodegradation of plastics. I also recommend rejecting the paper or major revision by better contextualizing these results.

Response 7: We agree with the reviewer’ comments and narrow the conclusive statements within a reasonable time-frame, 31 days. The current results showed that Enterobacter sp. D1 changed the physical morphology and chemical structure of the PE film, but it has not yet reached the entire “degradation”. As mentioned above, it was found that when the microorganisms were cultured with PE for a longer period of time, the PE film showed a greater degree of physical state change (data not shown) in our further studies. Additionally, we have completed genome sequencing of strain Enterobacter sp. D1 and found some degrading enzymes related to PE degradation, such as monooxygenase, dioxygenase, and P450 cytochrome oxidoreductase (data not shown) through functional annotation analysis. Meanwhile, we are also trying to further improve the degradation efficiency through other methods. Although there is still a distance from the entire “degradation”, Enterobacter sp. D1 has a significant PE-degrading capability compared to the very slow rate of PE degradation in the natural environment. Moreover, according to the literature that related to PE degradation in the recent years, it is still difficult to achieve complete degradation of PE film in the laboratory [3,7-9]. Nonetheless, it is much brighter than the degradation rate of PE in the natural environment.

 Reference

1.      Mehmood, C. T.; Qazi, I. A.; Hashmi, I.; Bhargava, S.; Deepa, S. Biodegradation of low density polyethylene (LDPE) modified with dye sensitized titania and starch blend using Stenotrophomonas pavanii. Int. Biodeterior. Biodegrad. 2016, 113, 276-286; DOI: 10.1016/j.ibiod.2016.01.025.

2.      Dang, T. C. H.; Nguyen, D. T.; Thai, H.; Nguyen, T. C.; Hien Tran, T. T.; Le, V. H.; Nguyen, V. H.; Tran, X. B.; Thao Pham, T. P.; Nguyen, T. G.; Nguyen, Q. T. Plastic degradation by thermophilic Bacillus sp. BCBT21 isolated from composting agricultural residual in Vietnam. Advances in Natural Sciences: Nanoscience and Nanotechnology. 2018, 9, (1), 015014; DOI: 10.1088/2043-6254/aaabaf.

3.      Yang, J.; Yang, Y.; Wu, W. M.; Zhao, J.; Jiang, L. Evidence of polyethylene biodegradation by bacterial strains from the guts of plastic-eating waxworms. Environ. Sci. Technol. 2014, 48, 13776-13784; DOI: 10.1021/es504038a.

4.      Da Costa, J. P.; Nunes, A. R.; Santos, P. S. M.; Girao, A. V.; Duarte, A. C.; Rocha-Santos, T. Degradation of polyethylene microplastics in seawater: Insights into the environmental degradation of polymers. Journal of environmental science and health. Part A, Toxic/hazardous substances & environmental engineering. 2018, 53, (9), 866-875; DOI: 10.1080/10934529.2018.1455381.

5.      Huerta Lwanga, E.; Thapa, B.; Yang, X.; Gertsen, H.; Salanki, T.; Geissen, V.; Garbeva, P. Decay of low-density polyethylene by bacteria extracted from earthworm's guts: A potential for soil restoration. The Science of the total environment. 2018, 624, 753-757; DOI: 10.1016/j.scitotenv.2017.12.144.

6.      Kyaw, B. M.; Champakalakshmi, R.; Sakharkar, M. K.; Lim, C. S.; Sakharkar, K. R. Biodegradation of Low Density Polythene (LDPE) by Pseudomonas Species. Indian J Microbiol. 2012, 52, 411-419; DOI: 10.1007/s12088-012-0250-6.

7.      Awasthi, S.; Srivastava, N.; Singh, T.; Tiwary, D.; Mishra, P. K. Biodegradation of thermally treated low density polyethylene by fungus Rhizopus oryzae NS 5. 3 Biotech. 2017, 7, (1), 73; DOI: 10.1007/s13205-017-0699-4.

8.      Kowalczyk, A.; Chyc, M.; Ryszka, P.; Latowski, D. Achromobacter xylosoxidans as a new microorganism strain colonizing high-density polyethylene as a key step to its biodegradation. Environ Sci Pollut Res Int. 2016, 23, 11349-11356; DOI 10.1007/s11356-016-6563-y.

9.      Shahnawaz, M.; Sangale, M. K.; Ade, A. B. Bacteria-based polythene degradation products: GC-MS analysis and toxicity testing. Environ Sci Pollut Res Int. 2016, 23, 10733-10741; DOI: 10.1007/s11356-016-6246-8.

Reviewer 2 Report

The article (ijerph-504713) entitled “Biodegradation of polyethylene by Enterobacter sp. D1 from the guts of wax moth Galleria mellonella” should be accepted with minor revisions.

The manuscript is very interesting and the scientific approach is good. Nowadays, it is important to develop methods to test the biodegradation of plastics from different origins and to improve the knowledge of PE-degrading microorganisms. The authors justify the study reporting good literature review and methods are well described; results and discussion are reported accordingly. My suggestion is only to improve the discussion, by adding references to other studies that tested PE degradation with different microorganisms. I think this might be useful for highlighting the importance of results obtained.

The conclusion are too long. I suggest to shift the literature references in the discussion section and to focus on the main results.

Author Response

Response to Reviewer 2 Comments

Point 1: The manuscript is very interesting and the scientific approach is good. Nowadays, it is important to develop methods to test the biodegradation of plastics from different origins and to improve the knowledge of PE-degrading microorganisms. The authors justify the study reporting good literature review and methods are well described; results and discussion are reported accordingly. My suggestion is only to improve the discussion, by adding references to other studies that tested PE degradation with different microorganisms. I think this might be useful for highlighting the importance of results obtained.

Response 1: The reviewer’s comments and suggestions are highly appreciated. Several studies on the degradation of PE by other species of microorganisms have been provided in the manuscript.

Point 2:  The conclusion are too long. I suggest to shift the literature references in the discussion section and to focus on the main results.

Response 2: As suggested, the conclusion has been modified accordingly.

Reviewer 3 Report

The manuscript focused on the biodegradation of polyethylene by Enterobacter sp. D1 from the guts of wax moth Galleria mellonella. The Authors were concentrated on isolation PE-degrading microorganisms from the guts of wax moth, the degradation efficiency as well as mechanisms determination.

Mechanisms determination were based on several methods such using SEM, AFM, FTIR, and LC-MS. However, in the manuscripts is not presented clearly mechanisms of the degradation process.

Some of my comments are posted below.

The aim of the research should contain the research hypothesis and highlight the novelty of the research work. What was the research hypothesis?

Line 22: arbonyl – should be karbonyl

The "Materials and methods section" should be changed and additional information is needed.

- source of origin of wax moth larvae,

- the sequences were submitted to the National Centre for Biotechnology Information - please add to the manuscript reference number of the strain

- description of biodegradation test

- why the biodegradation process was carried out during 31 day

Results and discussion

Figure 1 is poor quality, especially (c) and (d)

Figure 4 - the whole FTIR spectrum should be introduced, significant bands appear above 3000 cm-1

Section 3.2.3. Analysis of water-soluble products should be rewritten. The Authors detected monobenzyl phthalate - what was a source of this compound.

How would this compound formed? It is completely unclear and should be explained. Moreover, additional experiments should be done.

Results discussion should be improved.

More up to date references should be added.

Author Response

Response to Reviewer 3 Comments

Point 1: The aim of the research should contain the research hypothesis and highlight the novelty of the research work. What was the research hypothesis?

Response 1: According to the findings that polyethylene (PE) could be eaten by the wax moth by Paolo Bombelli et al [1], and the degradation of PE by YP1 and YTI derived from the gut of the Plodia interpunctella [2], our hypothesis is that there are some microorganisms which play a major role in the degradation process of PE in the wax moth.

In most studies, the screening of PE-degrading strains is mainly focusing on the natural environment sources such as ocean and land, while the screening of degradation strains from the gut of insects is less reported. Therefore, the novelty of our study is to screen microorganisms that can efficiently degrade PE from the gut of wax moth larvae.

Point 2: Line 22: arbonyl – should be karbonyl

Response 2: Thanks to the reviewers. According to the literature [2-6], the carbonyl is often expressed as “carbonyl” when it is used to represent the formation of surface function groups of PE film.

Point 3: source of origin of wax moth larvae

Response 3: The wax larvae were collected from the natural bee farms in Beijing and raised at Shanxi Agricultural University (with beeswax as the main food), which has also been provided in the manuscript. The beeswax, as a source of food for the wax moth, is a 'natural plastic' with a chemical structure similar to that of PE. Hence, the wax moth can be used to screen the microorganisms that “feed” PE.

Point 4: the sequences were submitted to the National Centre for Biotechnology Information - please add to the manuscript reference number of the strain

Response 4: As suggested, the GenBank accession number MK934326 has been added to the manuscript.

Point 5: description of biodegradation test

Response 5: We screened the strain D1 that could grow with 1% PE (one piece of PE film) as the sole carbon source both in the liquid and on the solid medium. Then strain D1 was inoculated in a liquid medium containing 5% PE (five pieces of PE film) as the sole carbon source to subject to the degradation test for 31-day. The corresponding content has been rewritten in the material and method section of the manuscript.

Point 6: why the biodegradation process was carried out during 31 day?

Response 6: The literature indicate that microbial degradation of PE is a relatively slow process under laboratory conditions. It is possible to observe the increases in ether and carboxyl functional groups when PE film is incubated with strain Stenotrophomonas pavanii (CC18) for 56 d [1], and it is possible that the properties and morphology of PE bags significantly changed following the treatment by Bacillus sp. BCBT21 for 30 d [2]. The biodegradation process is affected by many factors including microbial species, medium composition, culture environment and the structure of PE, therefore, we cannot predict whether the strain Enterobacter sp. D1 would degrade PE film before conducting the experiment. The selection of the 31-d duration of the experiment was mainly based on the previous study that the Enterobacter asburiae YT1 could cause obvious damage to the PE films over a 28-d incubation [3], and the steady changes of OD600 were observed when it was cultured for 31 days, along with the results that the physical and chemical properties of the PE film changed by the analysis of SEMAFM and FTIR in the current study. Certainly, the similar results may also be observed if the period of cultivation is 28 d or 35 d.

Point 7: Figure 1 is poor quality, especially (c) and (d)

Response 7: Figure 1 (c) and (d) have been replaced with the high quality graphs in the manuscript.

Point 8: Figure 4 - the whole FTIR spectrum should be introduced, significant bands appear above 3000 cm-1

Response 8: The whole FTIR spectrum has been introduced in the manuscript as suggested.

Point 9: Section 3.2.3. Analysis of water-soluble products should be rewritten. The Authors detected monobenzyl phthalate - what was a source of this compound.

Response 9: The reviewer’ suggestions are much appreciated. Analysis of water-soluble products has been rewritten in the manuscript.

Because of the unknownness of the PE degradation mechanism, we cannot simply analyze the source of monobenzyl phthalate according to the results of this study. Monobenzyl phthalate is the main metabolite of the butyl benzyl phthalate (BBP). BBP is a plasticizer added in the plastic manufacturing process. Monobenzyl phthalate was not detected in the control group in the current study, therefore, the production of this compound could be attributed to the action of Enterobacter sp.

Point 10: How would this compound formed? It is completely unclear and should be explained. Moreover, additional experiments should be done.

Response 10: Monobenzyl phthalate is the main metabolite of the plasticizer butyl benzyl phthalate (BBP). We speculate that the production of monobenzyl phthalate is caused by some oxidoreductases secreted by the strain Enterobacter sp. and the specific oxidation mechanism was further studied.

Currently, we have initially detected the production of alcohols, alkanes, hydrocarbon, esters and acids. The biodegradation mechanism of PE is complex and involves the participation of various oxidoreductases. Complete degradation of PE is not achieved, although some oxidoreductases such as laccase, monooxygenase and dioxygenase have been found to degrade PE [9-11]. In the present study, genome sequencing of strain Enterobacter sp. D1 was conducted and monooxygenase, dioxygenase, and P450 cytochrome oxidoreductase were found through functional annotation analysis (data not shown). We will attempt to treat PE with these enzymes to further analyze the relevant metabolites. Additionally, we will further conduct certain experiments to comprehensively analyze the degradation process. These include analysis of the molecular weight of the PE film by high-temperature gel permeation chromatography (HT-GPC), the weight loss of the PE film by gravimetric method and so on, which has also been stated in the manuscript.

Point 11: Results discussion should be improved. More up to date references should be added.

Response 11: As suggested, the results and discussion section have been improved, and more up to date references have been added.

Reference

1.      Bombelli, P.; Howe, C. J.; Bertocchini, F. Polyethylene bio-degradation by caterpillars of the wax moth Galleria mellonella. Curr Biol. 2017, 27, 292-293; DOI: 10.1016/j.cub.2017.02.060.

2.      Yang, J.; Yang, Y.; Wu, W. M.; Zhao, J.; Jiang, L. Evidence of polyethylene biodegradation by bacterial strains from the guts of plastic-eating waxworms. Environ. Sci. Technol. 2014, 48, 13776-13784; DOI: 10.1021/es504038a.

3.      Shahnawaz, M.; Sangale, M. K.; Ade, A. B. Bacteria-based polythene degradation products: GC-MS analysis and toxicity testing. Environ Sci Pollut Res Int. 2016, 23, 10733-10741; DOI: 10.1007/s11356-016-6246-8.

4.      Agamuthu, P.; Faizura, P. N. Biodegradability of degradable plastic waste. Waste Manage Res. 2005, 23, 95-100; DOI: 10.1177/0734242X05051045.

5.      Cai, L.; Wang, J.; Peng, J.; Wu, Z.; Tan, X., Observation of the degradation of three types of plastic pellets exposed to UV irradiation in three different environments. The Science of the total environment. 2018, 79, 740-747; DOI: 10.1016/j.scitotenv.2018.02.079.

6.      Orr, I. G.; Hadar, Y.; Sivan, A. Colonization, biofilm formation and biodegradation of polyethylene by a strain of Rhodococcus ruber. Appl Microbiol Biotechnol. 2004, 65, (1), 97-104. DOI: 10.1007/s00253-004-1584-8.

7.      Mehmood, C. T.; Qazi, I. A.; Hashmi, I.; Bhargava, S.; Deepa, S. Biodegradation of low density polyethylene (LDPE) modified with dye sensitized titania and starch blend using Stenotrophomonas pavanii. Int. Biodeterior. Biodegrad. 2016, 113, 276-286; DOI: 10.1016/j.ibiod.2016.01.025.

8.      Dang, T. C. H.; Nguyen, D. T.; Thai, H.; Nguyen, T. C.; Hien Tran, T. T.; Le, V. H.; Nguyen, V. H.; Tran, X. B.; Thao Pham, T. P.; Nguyen, T. G.; Nguyen, Q. T. Plastic degradation by thermophilic Bacillus sp. BCBT21 isolated from composting agricultural residual in Vietnam. Advances in Natural Sciences: Nanoscience and Nanotechnology. 2018, 9, (1), 015014; DOI: 10.1088/2043-6254/aaabaf.

9.      Santo, M.; Weitsman, R.; Sivan, A. The role of the copper-binding enzyme – laccase – in the biodegradation of polyethylene by the actinomycete Rhodococcus ruber. Int. Biodeterior. Biodegrad. 2012, 3, (1), 1-7. DOI: 10.1016/j.ibiod.2012.03.001.

10.    Chatterjee, S.; Roy, B.; Roy, D.; Banerjee, R. Enzyme-mediated biodegradation of heat treated commercial polyethylene by Staphylococcal species. Polym Degrad Stab. 2010, 95, (2), 195-200. DOI:
10.1016/j.polymdegradstab.2009.11.025.

11.     Devi, R.; Kannan, V.; Natarajan, K.; Nivas, D.; Kannan, K.; Chandru, S.; Antony, A. The Role of Microbes in Plastic Degradation. Environmental Waste Management. 2015, 341-370; DOI: 10.1201/b19243-13.

Round  2

Reviewer 1 Report

The authors have addressed the concerns I have. However, I think the experiment could be better designed and there needs to be a complete picture to better understand the results. The significance of the biodegradation in this experiment is low and I am unsure whether this process is scalable. I do recommend accepting the paper since the results look like a start.

Reviewer 3 Report

The Authors have provided appropriate changes, I do not have more questions.